# Artificial intelligence system reduces false-positive findings in the interpretation of breast ultrasound exams

Yiqiu Shen [1,6], Farah E. Shamout [2,6], Jamie R. Oliver[3,6], Jan Witowski [3], Kawshik Kannan[4], Jungkyu Park [5], Nan Wu[1], Connor Huddleston[3], Stacey Wolfson[3], Alexandra Millet[3], Robin Ehrenpreis[3], Divya Awal[3], Cathy Tyma[3], Naziya Samreen[3], Yiming Gao[3], Chloe Chhor[3], Stacey Gandhi[3], Cindy Lee[3], Sheila Kumari-Subaiya[3], Cindy Leonard[3], Reyhan Mohammed[3], Christopher Moczulski[3], Jaime Altabet[3], James Babb[3], Alana Lewin[3], Beatriu Reig [3], Linda Moy[3,5], Laura Heacock[3] & Krzysztof J. Geras [1,3,5 ✉]

Though consistently shown to detect mammographically occult cancers, breast ultrasound has been noted to have high false-positive rates. In this work, we present an AI system that achieves radiologist-level accuracy in identifying breast cancer in ultrasound images. Developed on 288,767 exams, consisting of 5,442,907 B-mode and Color Doppler images, the AI achieves an area under the receiver operating characteristic curve (AUROC) of 0.976 on a test set consisting of 44,755 exams. In a retrospective reader study, the AI achieves a higher AUROC than the average of ten board-certified breast radiologists (AUROC: 0.962 AI, 0.924 ± 0.02 radiologists). With the help of the AI, radiologists decrease their false positive rates by 37.3% and reduce requested biopsies by 27.8%, while maintaining the same level of sensitivity. This highlights the potential of AI in improving the accuracy, consistency, and efficiency of breast ultrasound diagnosis.

[1] Center for Data Science, New York University, New York, NY, USA. [2] Engineering Division, NYU Abu Dhabi, Abu Dhabi, UAE. [3] Department of Radiology, NYU Grossman School of Medicine, New York, NY, USA. [4] Department of Computer Science, Courant Institute, New York University, New York, NY, USA. [5] Vilcek Institute of Graduate Biomedical Sciences, NYU Grossman School of Medicine, New York, NY, USA. [6] These authors contributed equally: Yiqiu Shen, Farah E. Shamout, Jamie R. Oliver. ✉email: k.j.geras@nyu.edu

Breast cancer is the most frequently diagnosed cancer and the leading cause of cancer-related deaths among women worldwide[1]. It is estimated that 281,550 new cases of invasive breast cancer will be diagnosed among women in the United States in 2021, eventually leading to approximately 43,600 deaths[2]. Identifying breast cancer at an early stage before metastasis enables more effective treatments and therefore significantly improves survival rates[3,4]. Mammography has long been the most widely utilized imaging technique for screening and early detection of breast cancer, but it is not without limitations. In particular, for women with dense breast tissue, the sensitivity of mammography drops from 85% to 48–64%[5]. This is a significant drawback, as women with extremely dense breasts have a 4-fold increased risk of developing breast cancer[6]. Moreover, mammography is not always accessible, especially in limited-resources settings, where the high cost of equipment is prohibitive and skilled technologists and radiologists are not available[7].

Given the limitations of mammography, ultrasound (US) plays an important role in breast cancer diagnosis. It often serves as a supplementary modality to mammography in screening settings[8] and as the primary imaging modality in many diagnostic settings, including the evaluation of palpable breast abnormalities[9]. Moreover, US can help further evaluate and characterize breast masses and is therefore frequently used for performing image guided breast biopsies[10]. Breast US has several advantages compared to other imaging modalities, including relatively lower cost, lack of ionizing radiation, and the ability to evaluate images in real time[4]. In particular, US is especially effective at distinguishing solid breast masses from fluid-filled cystic lesions. In addition, breast US is able to detect cancers obscured on mammography, making it particularly useful in diagnosing cancers in women with mammographically dense breast tissue[11].

Despite these advantages, interpreting breast US is a challenging task. Radiologists evaluate US images using different features including lesion size, shape, margin, echogenicity, posterior acoustic features, and orientation, which vary significantly across patients[12]. Ultimately, they determine if the imaged findings are benign, need short-term follow-up imaging, or require a biopsy based on their suspicion of malignancy. There is considerable intra-reader variability in these recommendations and breast US has been criticized for increasing the number of false-positive findings[13,14]. Compared to mammography alone, the addition of US in breast cancer screening leads to an additional 5–15% of patients being recalled for further imaging and an additional 4–8% of patients undergoing biopsy[15–17]. However, only 7–8% of biopsies prompted by screening US are found to identify cancers[15,17].

Computer-aided diagnosis (CAD) systems have been proposed to assist radiologists in the interpretation of breast US exams over a decade ago[18]. Early CAD systems often relied on handcrafted visual features that are difficult to generalize across US images that were acquired using different protocols and US units[19–24]. Recent advances in deep learning have facilitated the development of AI systems for the automated diagnosis of breast cancer from US images[25–27]. However, the majority of these efforts rely on image-level or pixel-level labels, which require experts to manually mark images containing visible lesions within each exam or annotate lesions in each image, respectively[28–33]. As a result, existing studies have been based on small datasets consisting of several hundreds or thousands of US images. Deep learning models trained on those datasets might not sufficiently learn the diverse characteristics of US images observed in clinical practice. This is especially important for US imaging as lesion appearance can vary substantially depending on the imaging technique and the manufacturer of the US unit system. Moreover,

prior research has primarily focused on differentiating between benign and malignant breast lesions, hence evaluating AI systems only on the images which contain either benign or malignant lesions[34–36]. In contrast, the majority of breast cancer screening exams are negative (no lesions are present)[7,11]. In addition, most AI systems in previous studies do not interpret the model's predictions, resulting with "black-box" models[28–36]. So far, there has been little work on interpretable AI systems for breast US.

In this work, we present an AI system (Fig. 1) to identify malignant lesions in breast US images with the primary goal of reducing the frequency of false positive findings. In addition to classifying the images, the AI system also localizes the lesions in a weakly supervised manner[37–39]. That is, our AI system is able to explain its predictions by indicating locations of malignant lesions even though it is trained with binary breast-level cancer labels only (see Methods section 'Breast-level cancer labels'), which were automatically extracted from pathology reports. The explainability of our system enables clinicians to develop trust and better understand its strengths and limitations.

The proposed system provides several advances relative to previous work. First, to the best of our knowledge, the dataset used to train and evaluate this AI system is larger than any prior dataset used for this application[29,40]. Second, to understand the potential value of this AI system in clinical practice, we conducted a retrospective reader study to compare its diagnostic accuracy with ten board-certified breast radiologists. The AI system achieved a higher area under the receiver operating characteristic curve (AUROC) and area under the precision-recall curve (AUPRC) than the ten radiologists on average. Moreover, we showed that the hybrid model, which aggregates the predictions of the AI system and radiologists, improved radiologists' specificity and decreased biopsy rate while maintaining the same level of sensitivity. Of note, the term "prediction" refers to the diagnosis produced by AI/radiologists in this retrospective study as it is often used in the machine learning literature. It does not imply the study being prospective. In addition, we showed that the performance of the AI system remained robust across patients from different age groups and mammographic breast densities. Accuracy of our system also remained high when tested on an external data set[40].

## Results

**Datasets.** The AI system was developed and evaluated using the NYU Breast Ultrasound Dataset[41] consisting of 5,442,907 images within 288,767 breast US exams (including both screening and diagnostic exams) collected from 143,203 patients examined between 2012 and 2019 at NYU Langone Health in New York, USA. The NYU Langone hospital system spans multiple sites across New York City and Long Island, allowing the inclusion of a diverse patient population. The dataset included 28,914 exams associated with a pathology report, and among those, the biopsy or surgery yielded benign and malignant results for 26,843 and 5593 breasts, respectively. Patients in the dataset were randomly divided into a training set (60%) that was used for model training, a validation set (10%) that was used for hyperparameter tuning, and an internal test set (30%) that was used for model evaluation. Each patient was included in only one of the three sets. We used a subset of the internal test set for the reader study. The statistics of the overall dataset, the internal test set, and the reader study set are summarized in Table 1.

Each breast within an exam was assigned a label indicating the presence of cancer using pathology results. The pathology examinations were conducted on tissues obtained during a biopsy or breast surgery. As shown in Fig. 1b, all cancer-positive exams were accompanied by at least one pathology report

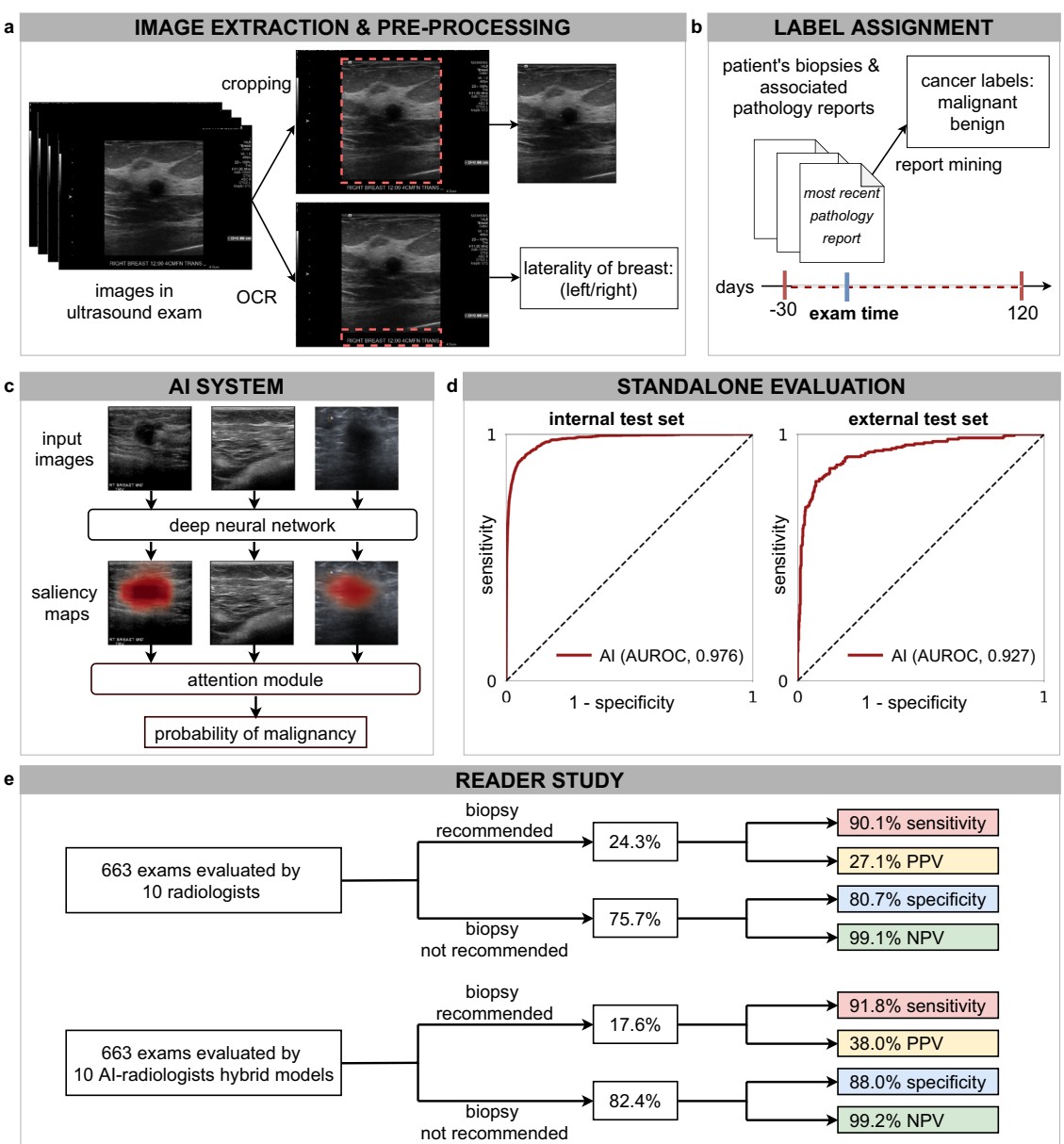

**Fig. 1 Overview of the system's pipeline. a** US images were pre-processed to extract the breast laterality (i.e., left or right breast) and to include only the part of the image which shows the breast (cropping out the image periphery which typically contains textual metadeta about the patient and US acquisition technique). **b** For each breast, we assigned a cancer label using the recorded pathology reports for the respective patient within −30 and 120 days from the time of the US examination. We applied additional filtering on the internal test set to ensure that cancers in positive exams are visible in the US images and negative exams have at least one cancer-negative follow-up (see Methods section `Additional filtering of the test'). **c** The AI system processes all US images acquired from one breast to compute probabilistic predictions for the presence of malignant lesions. The AI system also generates saliency maps that indicate the informative regions in each image. **d** We evaluated the system on an internal test set (AUROC: 0.976, 95% CI: 0.972, 0.980, $n = 79,156$ breasts) and an external test set (AUROC: 0.927, 95% CI: 0.907, 0.959, $n = 780$ images). **e** In a reader study consisting of 663 exams ($n = 1024$ breasts), we showed that the AI system can improve the specificity and positive predictive value (PPV) for 10 attending radiologists while maintaining the same level of sensitivity and negative predictive value (NPV).

indicating malignancy collected either 30 days prior or 120 days after the US examination. This time frame was chosen to maximize the inclusion of both lesions found at primary screening US and lesions found during targeted US after an initial imaging workup with a different modality. We filtered the internal test set to ensure that cancers were visible on positive exams and that negative exams had either cancer-negative biopsy or at least one negative follow-up US exam (see Methods section 'Additional filtering of the test set'). Studies with neither a pathology report nor any negative follow-up were included in

the training and validation set but excluded from the internal test set.

To assess the ability of the AI system to generalize across patient populations and image acquisition protocols, we further evaluated it on the public Breast Ultrasound Images (BUSI) dataset collected at the Hospital for Early Detection and Treatment of Women's Cancer in Cairo, Egypt[40]. This external test set consisted of 780 images, of which 437 were benign, 210 were malignant, and 133 were negative (no lesion present). These images were collected from 600 patients. Of note, the BUSI

Table 1 Statistics of the overall NYU Breast Ultrasound Dataset, internal test set, and reader study set. This dataset was collected from NYU Langone Health over an eight-year period. Exam-level BI-RADS were issued by radiologists based on patients' breast US exams. Breast densities were determined using existing screening and diagnostic mammography reports. Patients who were not matched with any mammograms were assigned "unknown" for breast density. Abbreviations: N, number; SD, standard deviation.

| Characteristic, unit | Overall | Internal test set | Reader study |
|---|---|---|---|
| Patients, N | 143,203 | 25,003 | 644 |
| Age, mean years (SD) | 53.7 (13.7) | 55.5 (12.7) | 52.8 (14.0) |
| < 40 yrs old, N (%) | 18,218 (12.7) | 1857 (7.4) | 90 (14.0) |
| 40 − 49 years old, N (%) | 33,955 (23.7) | 5811 (23.2) | 175 (27.2) |
| 50 − 59 years old, N (%) | 34,942 (24.4) | 6567 (26.3) | 146 (22.7) |
| 60 − 69 years old, N (%) | 26,671 (18.6) | 5198 (20.8) | 104 (16.1) |
| ≥70 years old, N (%) | 17,703 (12.4) | 3359 (13.4) | 81 (12.6) |
| Exams, N | 288,767 | 44,755 | 663 |
| Images, N | 5,442,907 | 858,636 | 13,582 |
| Average no. of images per exam, N | 18 | 19 | 20 |
| Exams associated with biopsy, N (%) | 28,914 (10.0) | 8337 (18.6) | 587 (88.5) |
| Breasts, N | 510,271 | 79,156 | 1024 |
| Breasts with benign findings, N | 26,843 | 7879 | 567 |
| Breasts with malignant findings, N | 5593 | 1324 | 73 |
| Exam-level BI-RADS | | | |
| BI-RADS 0, N (%) | 14,078 (4.9) | 1092 (2.4) | 80 (12.1) |
| BI-RADS 1, N (%) | 86,347 (29.9) | 12,374 (27.6) | 56 (8.4) |
| BI-RADS 2, N (%) | 136,322 (47.2) | 21,675 (48.4) | 80 (12.1) |
| BI-RADS 3, N (%) | 27,711 (9.6) | 3586 (8.0) | 25 (3.8) |
| BI-RADS 4, N (%) | 22,133 (7.7) | 5578 (12.5) | 391 (59.0) |
| BI-RADS 5, N (%) | 1348 (0.5) | 338 (0.8) | 22 (3.3) |
| BI-RADS 6, N (%) | 518 (0.2) | 69 (0.2) | 3 (0.5) |
| Unknown BI-RADS, N (%) | 310 (0.1) | 43 (0.1) | 6 (0.9) |
| Exam-level mammographic density | | | |
| A (breasts are almost entirely fatty), N (%) | 5384 (1.9) | 695 (1.6) | 13 (2.0) |
| B (scattered areas of fibroglandular density), N (%) | 69,948 (24.2) | 11,048 (24.7) | 143 (21.6) |
| C (breasts are heterogeneously dense), N (%) | 165,855 (57.4) | 26,509 (59.2) | 376 (56.7) |
| D (the breasts are extremely dense), N (%) | 31,829 (11.0) | 5189 (11.6) | 76 (11.5) |
| Unknown density, N (%) | 15,751 (5.5) | 1314 (2.9) | 55 (8.3) |

dataset was acquired using different US machines and was collected from patients with contrasting demographic backgrounds compared to the NYU dataset. Each image in the BUSI dataset was associated with a label indicating the presence of any malignant lesions.

**AI system performance**. On the internal test set of 44,755 US exams (25,003 patients, 79,156 breasts), the AI system achieved an AUROC of 0.976 (95% CI: 0.972, 0.980) in identifying breasts with malignant lesions. Additionally, we stratified patients by age, mammographic breast density, US machine manufacturer, and evaluated AI model performance across these sub-populations (Table 2). The AI system maintained high diagnostic accuracy among all age groups (AUROC: 0.969–0.981), mammographic breast densities (AUROC: 0.964–0.979), and US device manufacturers (AUROC: 0.974–0.990). We also explored the impact of training dataset size on the performance of AI system. We observed that more training data led to a better AUROC (Supplementary Table 1). In addition, we evaluated the AI system on the external test set (BUSI dataset)[40]. Even though the AI system was not trained on any images of the external test set, it maintained a high level of diagnostic accuracy (0.927 AUROC, 95% CI: 0.907, 0.959).

**Reader study**. To compare the performance of the AI system with that of breast radiologists, we conducted two reader studies: one on the internal test set and the other on the external BUSI dataset. Conclusions drawn from the results for both datasets were

consistent. Here we present the results for the internal test set. The results for the external test set are in the Supplementary Information.

From the internal test set, we constructed a reader study subset by selecting 663 exams (644 patients, 1024 breasts). Among the exams selected for this study, 73 breasts had pathology-proven cancer, 535 breasts had a biopsy yielding exclusively benign findings, and 416 breasts were not biopsied but were evaluated by radiologists as likely benign and had a follow-up benign evaluation at 1–2 years. Readers were informed that the study dataset was enriched with cancers but were not informed of the enrichment level.

Ten board-certified breast radiologists rated each breast according to the Breast Imaging Reporting and Data System (BI-RADS)[12]. Radiologists' experience is described in Supplementary Table 2. Readers were provided with contextual information typically available in the clinical setting, including the patient's age, burnt-in annotations showing measurements of suspicious findings, and notes from the technologist, such as specifying any region of palpable concern or pain. In contrast, the AI system was not provided any contextual information.

For each reader, we computed a receiver operating characteristic (ROC) curve and a precision-recall curve by comparing their BI-RADS scores to the ground-truth outcomes (see Methods section 'Statistical analysis'). The ten radiologists achieved an average AUROC of 0.924 (SD: 0.020, 95% CI: 0.905, 0.944) and an average AUPRC of 0.565 (SD: 0.072, 95% CI: 0.465, 0.625) (Supplementary Figure 1). Compared to the average radiologist in this study, the AI system achieved a higher AUROC of 0.962

**Table 2 AI performance on the internal test set across different sub-populations.** We reported the AUROC of the AI system with 95% confidence intervals on the internal test set. The biopsied population only includes exams where at least one biopsy was recommended. We stratified exams based on patient age, mammographic breast density, and the manufacturer of the US devices. Mammographic breast density was categorized based on the BI-RADS standards[69].

| Population | AUROC (95% CI) | No. of breasts | No. of cancers |
|---|---|---|---|
| Overall population | 0.976 (0.972, 0.980) | 79,078 | 1248 |
| Biopsied population | 0.940 (0.934, 0.947) | 12,973 | 1248 |
| Age | | | |
| < 40 yrs old | 0.969 (0.955, 0.982) | 5176 | 72 |
| 40 − 49 yrs old | 0.970 (0.955, 0.986) | 19,677 | 160 |
| 50 − 59 yrs old | 0.981 (0.975, 0.986) | 24,142 | 292 |
| 60 − 69 yrs old | 0.980 (0.973, 0.985) | 19,039 | 326 |
| ≥70 yrs old | 0.969 (0.958, 0.981) | 11,044 | 398 |
| Breast density | | | |
| Entirely fatty | 0.964 (0.942, 0.983) | 1157 | 54 |
| Scattered fibroglandular densities | 0.975 (0.961, 0.982) | 19,199 | 441 |
| Heterogeneously dense | 0.979 (0.974, 0.981) | 47,255 | 610 |
| Extremely dense | 0.964 (0.932, 0.973) | 9398 | 90 |
| Unkown | 0.970 (0.955, 0.983) | 2069 | 53 |
| Manufacturer | | | |
| GE | 0.984 (0.968, 0.993) | 5708 | 47 |
| Medison | 0.990 (0.974, 0.996) | 2673 | 13 |
| Philips | 0.977 (0.970, 0.982) | 28,943 | 412 |
| Siemens | 0.974 (0.968, 0.980) | 37,572 | 699 |
| Toshiba | 0.986 (0.978, 0.992) | 4180 | 77 |
| Other | — | 2 | 0 |

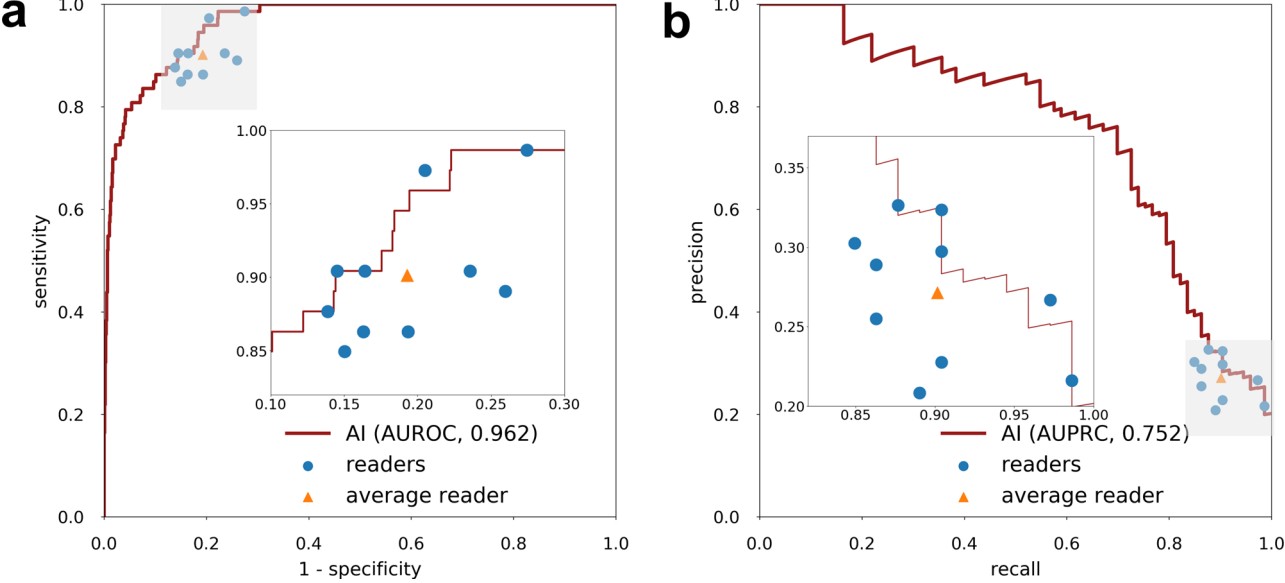

**Fig. 2 Reader study results.** The performance of the AI system on the reader study population ($n = 1024$ breasts) using ROC curve (**a**) and precision-recall curve (**b**). The AI achieved 0.962 (95% CI: 0.943, 0.979) AUROC and 0.752 (95% CI: 0.675, 0.849) AUPRC. Each data point represents a single reader and the triangles correspond to the average reader performance. The inset shows a magnification of the gray shaded region.

(95% CI: 0.943, 0.979) with an AUROC improvement of 0.038 (95% CI: 0.028, 0.052, $P < 0.001$) and a higher AUPRC of 0.752 (95% CI: 0.675, 0.849) with an AUPRC improvement of 0.187 (95% CI: 0.140, 0.256, $P < 0.001$) (Fig. 2). In addition, we also compared the specificity and sensitivity achieved by the AI system and radiologists. We assigned a positive prediction to any breast a radiologist gave a BI-RADS score of ≥4, and a negative prediction to any breast that was given a BI-RADS score of 1–3. A BI-RADS score of ≥4 is an assessment that indicates a radiologist thinks an exam is suspicious for malignancy. This was selected as the threshold for positive predictions since this is the score above which a patient will typically undergo an invasive procedure

(biopsy or surgical excision) to definitively determine whether they have cancer[12]. With this methodology, the ten radiologists achieved an average specificity of 80.7% (SD: 4.7%, 95% CI: 78.9%, 82.6%) and an average sensitivity of 90.1% (SD: 4.3%, 95% CI: 86.4%, 93.8%). At the average radiologist's specificity, the AI system achieved a sensitivity of 94.5% (95% CI: 89.4%, 100.0%) and an improvement in sensitivity of 4.4% (95% CI: −0.3%, 7.5%, $P = 0.0278$). At the average radiologist's sensitivity, the AI system achieved a higher specificity of 85.6% (95% CI: 83.9%, 88.0%) with an absolute increase in specificity of 4.9% (95% CI: 3.0%, 7.1%; $P < 0.001$). At the average radiologist's sensitivity, the AI system recommended tissue biopsies on 19.8% (95% CI: 17.9%,

22.1%) of breasts and 32.5% (95% CI: 26.9%, 39.2%) of these biopsies were for breasts ultimately found to have cancer. Compared to the average reader's biopsy rate of 24.3% (SD: 4.5%, 95% CI: 22.0%, 26.5%) and average PPV of 27.1% (SD: 4.1%, 95% CI: 22.9%, 33.1%), the AI system achieved an absolute reduction in biopsy rate of 4.5% (95% CI: 2.9%, 6.5%, $P < 0.001$) which corresponds to 18.6% of all biopsies recommended by the average radiologist and achieved an absolute improvement in PPV of 5.4% (95% CI: 2.4%, 8.9%, $P < 0.001$). The performance of the AI system and readers is summarized in Supplementary Table 3.

**Subgroup analysis on the biopsied population.** We conducted additional analyses on two clinically relevant subgroups in the reader study to understand the relative strengths of the AI system and radiologists. The first analysis examined diagnostic accuracy exclusively amongst breasts with lesions that had undergone biopsy or surgical evaluation (73 breasts with pathology-confirmed malignant lesions and 535 breasts with exclusively pathology-confirmed benign lesions). Breasts that yielded normal findings were not included. As expected, compared to the overall reader study population, AUROC (mean: 0.896, SD: 0.024, 95% CI: 0.874, 0.929) and specificity (mean: 69.8%, SD: 6.9%, 95% CI: 67.7%, 73.6%) of radiologists declined in this sub-population. Additionally, the average biopsy rate of radiologists increased to 37.4% (SD: 6.4%, 95% CI: 33.1%, 39.8%). On this subgroup, the AI system achieved an AUROC of 0.941 (95% CI: 0.922, 0.968). Compared to radiologists, the AI system demonstrated an absolute improvement of 8.5% (95% CI: 5.3%, 11.1%; $P < 0.001$) in specificity, an absolute reduction of 7.5% (95% CI: 4.4%, 9.6%, $P < 0.001$) in biopsy rate, and an absolute improvement in PPV of 6.7% (95% CI: 3.0%, 9.8%, $P < 0.001$), while matching the average radiologist's sensitivity. The performance of each reader is shown in Supplementary Table 4.

Next, we evaluated the accuracy of readers and the AI system exclusively amongst breasts with pathology-confirmed cancers (97 malignant lesions across 73 breasts). As shown in Supplementary Table 5, we stratified malignant lesions by cancer subtype, histologic grade, and biomarker profile. This was done to further investigate the AI system's ability to discriminate between benign and malignant lesions. Certain types of breast cancers (such as high grade, triple biomarker negative cancers) may closely resemble benign masses (more likely to have oval/round shape and circumscribed margins, less likely to have posterior attenuation compared to other cancers) and are considered particularly difficult to characterize[42]. Although the sample sizes in some subgroups are limited, this analysis demonstrated that the sensitivity of the AI system was similar to that of the readers across all stratification categories. There were no significant differences in sub-populations of patients where the AI system had inferior performance.

**Qualitative analysis of saliency maps.** In an attempt to understand the AI system's potential utility as a decision support tool, we qualitatively assessed six studies using the AI's saliency maps. These saliency maps indicated where the system identified potentially benign and malignant lesions, and represent data that could be made available to radiologists (in addition to breast level predictions of malignancy) if the AI system were integrated into clinical practice. Figure 3a, b shows two 1.5cm irregularly shaped hypoechoic masses with indistinct margins, that ultimately underwent biopsy and were found to be invasive ductal carcinoma. All readers as well as the AI system correctly identified these lesions as being suspicious for malignancy. Figure 3c displays a small 7mm complicated cystic/solid nodule with a

microlobulated contour, which 7 out of 10 readers as well as the AI system thought appeared benign. However, this lesion ultimately underwent biopsy and was found to be invasive ductal carcinoma. Figure 3d displays a 7mm superficial and palpable hypoechoic mass with surrounding echogenicity, that underwent biopsy and was found to be benign fat necrosis. However, the AI system as well as 9 out of 10 readers incorrectly thought this lesion was suspicious for malignancy, and recommended it undergo biopsy. Lastly, Fig. 3e shows a small 7mm ill-defined area and Fig. 3f displays a 9mm mildly heterogenous lobulated solid nodule. All 10 radiologists thought these two lesions appeared suspicious and recommended they undergo biopsy. In contrast, the AI system correctly classified the exams as benign, and the lesions were ultimately found to be benign fibrofatty tissue (Fig. 3e) and a fibroadenoma (Fig. 3f). Although we were unable to determine clear patterns among these US exams, the presence of cases where the AI system correctly contradicted the majority of readers and produced appropriate localization information underscores the potential complementary role the AI system might play in helping human readers more frequently reach accurate diagnoses. We provided additional visualization of saliency maps in Supplementary Fig. 2.

**Potential clinical applications.** To evaluate the potential of our AI system to augment radiologists' diagnosis, we created hybrid models of the AI system and the readers. The predictions of each hybrid model were computed as an equally weighted average between the AI system and each reader (see Methods section 'Hybrid model'). This analysis revealed that the performance of all readers was improved by incorporating the predictions of the AI system (Fig. 4, Supplementary Table 6). On average, the hybrid models improved radiologists' AUROC by 0.037 (SD: 0.013, 95% CI: 0.011, 0.070, $P < 0.001$) and improved their AUPRC by 0.219 (SD: 0.060, 95% CI: 0.089, 0.372, $P < 0.001$). At the radiologists' sensitivity levels, the hybrid models increased their average specificity from 80.7% to 88.0% (average increase 7.3%, SD: 3.8%, 95% CI: 2.7%, 18.5%, $P < 0.001$), increased their PPV from 27.1% to 38.0% (average increase 10.8%, SD: 5.3%, 95% CI: 3.7%, 25.0%, $P < 0.001$), and decreased their average biopsy rate from 24.3% to 17.6% (average decrease 6.8%, SD: 3.5%, 95% CI: 2.3%, 17.1%, $P < 0.001$). The reduction in biopsies achieved by the hybrid model represented 27.8% of all biopsies recommended by radiologists. In Supplementary Table 7, we reported the numbers of false positive biopsies (FP) and false negative diagnoses (FN) of all ten radiologists and hybrid models. We divided FP and FN according to the BI-RADS scores given by each radiologist. On average, the radiologists made 182.7 (SD: 43.8, 95% CI: 178.8, 185.9) FPs and 53.3 (SD: 18.4, 95% CI: 52.4, 54.3) of them were rated as BI-RADS 4A by the readers. The hybrid models decreased average radiologists' FPs by 37.3% (SD: 12.9%, 95% CI: 35.5%, 39.3%, $P<0.001$) while yielding the same number or fewer FN than the radiologists. In particular, 68.2% (SD: 19.5%, 95% CI: 65.3%, 70.1%) of radiologists' FPs in BI-RADS 4A category were obviated.

In addition, the AI system could also be used to assist radiologists to triage US exams (Supplementary Table 8). To evaluate the potential of the AI system in identifying cancer-negative cases with high confidence, we selected a very low decision threshold to triage women into a no-radiologist work stream. On the reader study subset, using this triage paradigm, the AI system achieved an NPV of 99.86% while retaining a specificity of 77.7%. This result suggests that it may be feasible to dismiss 77.7% of normal/benign cases and skip radiologist review if we accept missing one cancer in every 740 negative predictions, which is less than 1/6 of the false negative rate observed among

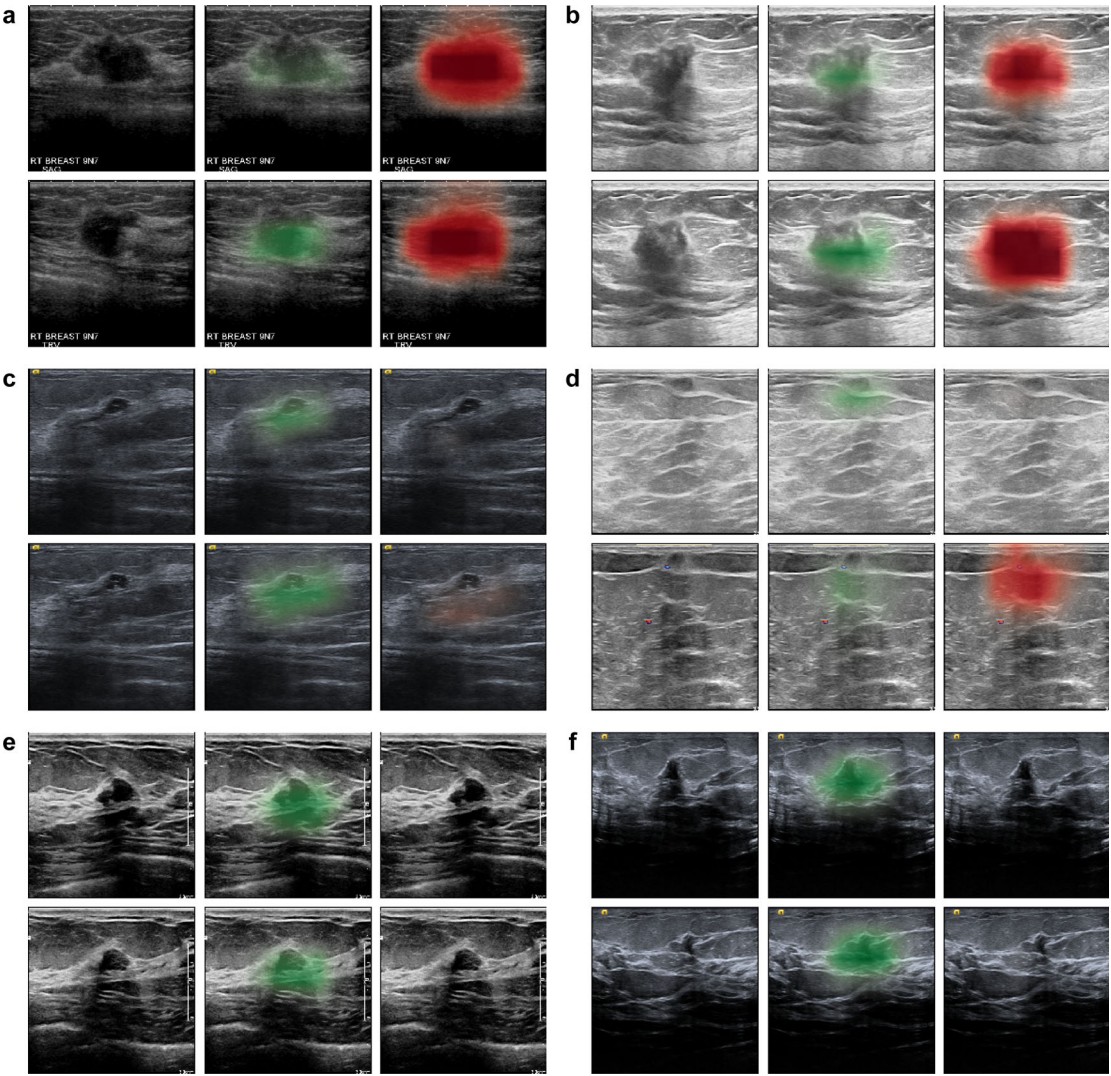

**Fig. 3 Qualitative analysis of saliency maps.** In each of the six cases (**a**–**f**) from the reader study, we visualized the sagittal and transverse views of the lesion (left) and the AI's saliency maps indicating the predicted locations of benign (middle) and malignant (right) findings (see Methods section `Deep neural network architecture'). Exams **a**–**c** display lesions that were ultimately biopsied and found to be malignant. All readers and the AI system correctly classified exams **a**–**b** as suspicious for malignancy. However, the majority of readers (7/10) and the AI system incorrectly classified case **c** as benign. Cases **d**–**f** display lesions that were biopsied and found to be benign. The majority of readers incorrectly classified exams **d** (9/10), **e** (10/10), and **f** (10/10) as suspicious for malignancy and recommended the lesions undergo biopsy. In contrast, the AI system classified exam **d** as malignant, but correctly identified exams **e**–**f** as being benign.

radiologists in the reader study (one missed cancer for every 109 negative evaluations). To evaluate the potential of the AI system in triaging patients into an enhanced assessment work stream, we used a very high decision threshold. In this enhanced assessment work stream, the AI system achieved a PPV of 84.4% while retaining a sensitivity of 52.1%. These results suggest that it may be feasible to rapidly prioritize more than half of cancer cases, with approximately five out of six biopsies leading to a diagnosis of cancer. For comparison, only 27.1% biopsies that the radiologists recommended were diagnosed with cancer. While we demonstrated the potential of AI in automatically triaging breast US exams, confirmation of these performance estimates would require extensive validation in a clinical setting.

## Discussion

In this work, we present a radiologist-level AI system that is capable of automatically identifying malignant lesions in breast US images. Trained and evaluated on a large dataset collected from 20 imaging sites affiliated with a large medical center, the AI system maintained a high level of diagnostic accuracy across a diverse range of patients whose images were acquired using a variety of US units. By validating its performance on an external dataset, we produced preliminary results substantiating its ability to generalize across a patient cohort with different demographic composition and image acquisition protocols.

Our study has several strengths. First, in the reader study subset, we found that the AI system performed comparably to board-certified breast radiologists. The ten radiologists achieved an average sensitivity of 90.1% (SD: 4.3%, 95% CI: 86.4%, 93.8%) and an average specificity of 80.7% (SD: 4.7%, 95% CI: 78.9%, 82.6%). The sensitivity of radiologists in our study is consistent with the results reported in other breast US reader studies[10,43], as well as the sensitivity of breast radiologists observed in clinical practice, despite the fact that radiologists in our study did not have access to the patient's medical record or prior breast

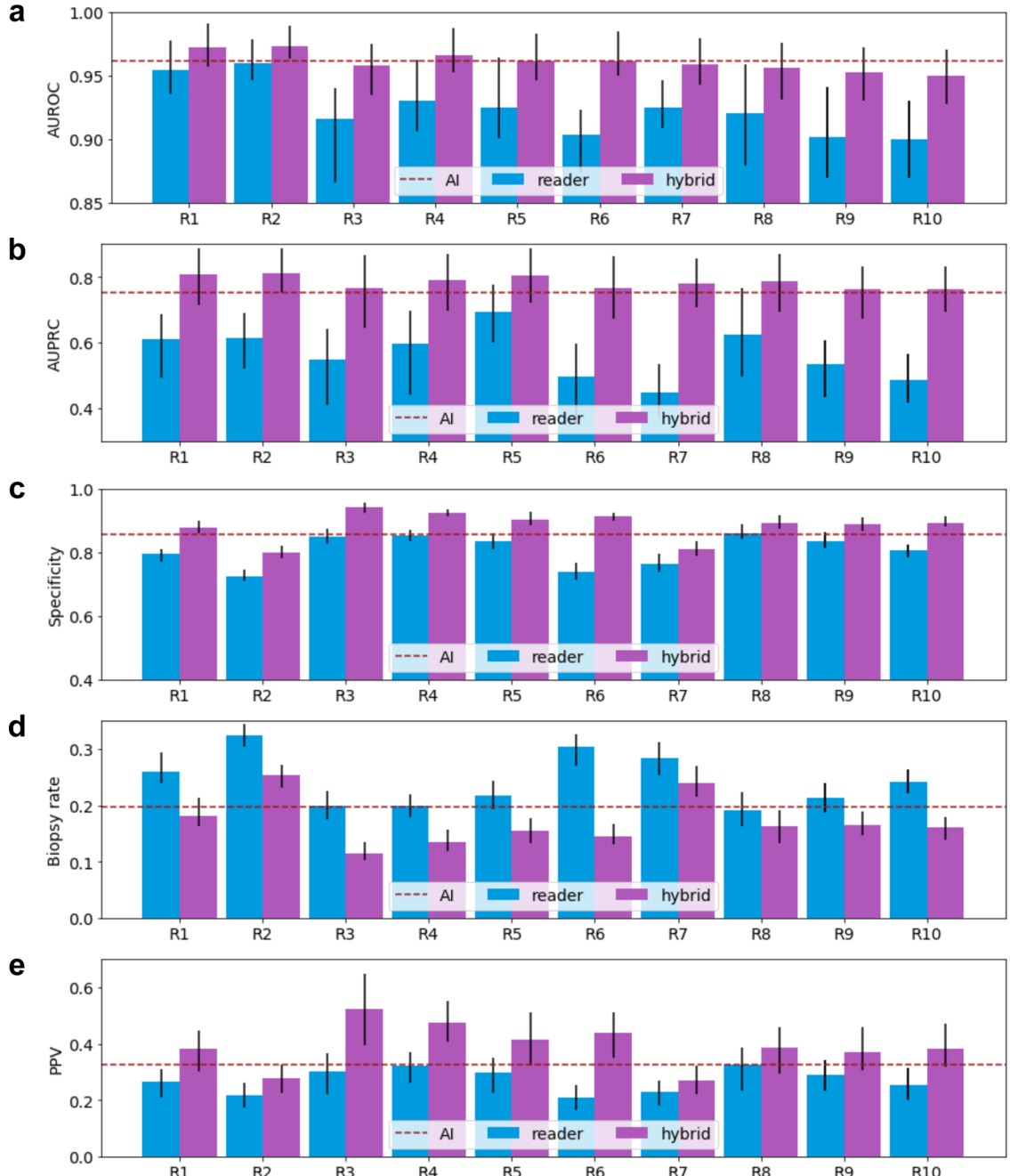

**Fig. 4 Performance of readers, AI, and hybrid models.** We reported the observed values (measure of center) and 95% confidence intervals (error bars) of AUROC (**a**), AUPRC (**b**), specificity (**c**), biopsy rate (**d**), and PPV (**e**) of ten radiologists (R1-R10), AI, and the hybrid models on the reader study set ($n =$ 1024 breasts) The predictions of each hybrid model are weighted averages of each reader's BI-RADS scores and the AI's probablistic predictions (see Methods section `Hybrid model'). We dichotomized each hybrid model's probabilistic predictions to match the sensitivity of its respective reader. We dichotomized the AI's predictions to match the average radiologists' sensitivity. The collaboration between AI and readers improves readers' AUROC, AUPRC, specificity, and PPV, while reducing biopsy rate. We estimated the 95% confidence intervals by 1000 iterations of the bootstrap method.

imaging[15,44]. Compared to radiologists in our reader study, the AI system was able to detect cancers with the same sensitivity, while obtaining a higher specificity (85.6%, 95% CI: 83.9%, 88.0%), a higher PPV (32.5%, 95% CI: 26.9%, 39.2%), and a lower biopsy rate (19.8%, 95% CI: 17.9%, 22.1%). Moreover, the AI system achieved a higher AUROC (0.962, 95% CI: 0.943, 0.979) and AUPRC (0.752, 95% CI: 0.675, 0.849) than all ten radiologists. This trend was confirmed in our subgroup analysis which showed that the system could accurately interpret US exams that are deemed difficult by radiologists.

Another strength of this study is that we explored the benefits of collaboration between radiologists and AI. We proposed and evaluated a hybrid diagnostic model that combined the predictions from radiologists and the AI system. The results from our reader study suggest that such collaboration improves the diagnostic accuracy and reduces false positive biopsies for all ten radiologists (Supplementary Table 6). In fact, breast US has come under criticism for having a high false positive rate[13,14]. As reported by multiple clinical studies, only 7-8% of breast biopsies performed under US guidance are found to yield cancers[15,17].

Indeed, for the ten radiologists in our cancer-enriched reader study subset, on average 19.3% (SD: 4.7%, 95% CI: 17.7%, 20.6%) of cancer-negative exams were falsely diagnosed as positive and only 27.1% (SD: 4.1%, 95% CI: 22.9%, 33.1%) of the exams that they recommended to undergo biopsy actually had cancer. In this study, we showed that the hybrid models reduced the average radiologist's false positive rate to 12.0 (SD: 3.9%, 95% CI: 7.6%, 21.0%), representing a 37.3% (SD: 12.9%, 95% CI: 35.5%, 39.3%) relative reduction. The hybrid models also increased the average radiologist's PPV to 38.0% (SD: 6.0%, 95% CI: 24.1%, 50.0%). These results indicate that our AI system has the potential to aid radiologists in their interpretation of breast US exams to reduce the number of false positive interpretations and benign biopsies performed.

Beyond improving radiologists' performance, we also explored how AI systems could be utilized to assist radiologists to triage US exams. We showed that high-confidence operating points provided by the AI system can be used to automatically dismiss the majority of low-risk benign exams and escalate high-risk cases to an enhanced assessment stream. Prospective clinical studies will be required to understand the full extent to which this technology can benefit US reading.

Finally, we have made technical contributions to the methodology of deep learning for medical image analysis. Prior work on AI systems for interpreting breast US exams, and other similar applications, rely on manually collected image-level or pixel-level labels[28-33]. In contrast, our AI system was trained using breast-level labels which were automatically extracted from pathology reports. This is an important difference, as developing a reliable AI system for clinical use requires training and validation on large-scale datasets to ensure the network will function well across the broad spectrum of cases encountered in clinical practice. At such a scale, it is impractical to collect labels manually. We address this issue by adopting the weakly supervised learning paradigm to train models at scale without the need for image-level or pixel-level labels. This paradigm enables the model to generate interpretable saliency maps that highlight informative regions in each image. Admittedly, the literature has not yet reached a consensus on the definition of what exactly interpretability for neural networks is. Nevertheless, with the saliency maps, researchers can perform qualitative error analysis and understand the strength and limitations of the AI system. Furthermore, a system trained with such a large dataset could help discover novel data-driven imaging biomarkers, leading to a better understanding of breast cancer.

Despite the contributions of our study in advancing breast cancer diagnosis, it has some limitations. We focused on the evaluation of an AI system that detects breast cancer only using US imaging. In clinical practice, US imaging is often used as a complementary modality to mammography. One promising research direction is to utilize multimodal learning[45,46] to combine information from other imaging modalities. Moreover, the diagnosis produced by our AI system is based only on a single US exam, while breast radiologists often refer to patients' prior imaging to evaluate the morphological changes of suspicious findings over time. Future research could focus on augmenting AI systems to extract relevant information from past US exams. In addition, we did not provide an evaluation on patient cohorts stratified by risk factors such as family history of breast cancer and BRCA gene test results.

Another limitation of this work is the design of reader study. To provide a fair comparison with the AI system, readers in our study were only provided with US images, patients' ages, and notes from the operating technician. In clinical practice, breast radiologists also have access to other information such as patients' prior breast imaging and their electronic medical

records. Moreover, in the breast cancer screening setting, a screening US examination is typically accompanied by a screening mammogram. Even if prior US exams are not available, radiologists can typically refer to the mammogram for additional information, which can also influence the way that an US exam is interpreted. In addition, the qualitative analysis presented in this study was conducted over a limited set of exams. A systematic study on the differences between the AI system and the perception of radiologists in sonography interpretation is required to understand the limitations of such systems.

Finally, compared to the NYU Breast Ultrasound Dataset, the external test set is limited in size. All images in the external test set were acquired using a single US system[40]. Moreover, each lesion/finding in the external test set is only associated with a single image. On the contrary, in clinical practice, the technicians often acquire multiple images from different views for findings that are suspicious of malignancy. This difference in image acquisition protocol likely lead to the gap in AI's performance between the internal and external test set.

Despite these limitations, we believe this study is a meaningful contribution to the emerging field of AI-based decision support systems for interpreting breast US exams. On a clinically realistic population, our AI system achieved a higher diagnostic accuracy (AUROC: 0.976, 95% CI: 0.972, 0.980) than prior AI systems for breast US lesion classification (AUROC: 0.82-0.96)[32,34,47-52], though we acknowledge these systems can be compared only approximately as they were evaluated on different datasets. Key features that contributed to our AI system's high level performance were the large dataset used in training, along with utilization of the weakly supervised learning paradigm that enables the system to learn from automatically extracted labels. Furthermore, as our AI system was evaluated on a large test set (>44,000 US exams) acquired from a diverse range of US units and patients of diverse demographics, we are optimistic of its ability to perform well prospectively, in the hands of radiologists. A few recent studies have demonstrated in retrospective reader studies that AI systems can improve the performance of radiologists when they have access to the decision support tool while reviewing US exams[47,51]. However, these studies utilized an AI system that required radiologists to localize lesions by manually drawing bounding boxes. Moreover, these studies used small datasets and did not evaluate the AI's performance on sub-populations stratified by age and breast density. This makes it hard to determine if the system would maintain performance across the broad range of US exams that a radiologist might encounter in different clinical settings. Regardless of these limitations, these studies demonstrate that an AI system with a relatively low AUROC of 0.86-0.88 can substantially improve the diagnostic accuracy of radiologists. Based on these results, we are optimistic that our AI system, which does not require radiologists to localize lesions and achieved a higher diagnostic accuracy (AUROC: 0.976) on a larger diverse patient population, could enable radiologists to achieve even greater levels of performance. As a next step, our system requires prospective validation before it can be widely deployed in clinical practice. The potential impact that such a system could have on women's imaging is immense, given the enormous volume of women who undergo breast US exams each year.

In conclusion, we examined the potential of AI in US exam evaluation. We demonstrated in a reader study that deep learning models trained with a sufficiently large amount of data are able to produce diagnosis as accurate as experienced radiologists. We further showed that the collaboration between AI and radiologists can significantly improve their specificity and obviate 27.8% of requested biopsies. We believe this research could supplement future approaches to breast cancer diagnosis. In addition,

the general approach employed in our work, mainly the framework for weakly supervised classification and localization, may enable utilization of deep learning in similar medical image analysis tasks.

## Methods

**Ethical approval.** This retrospective study was approved by the NYU Langone Health Institutional Review Board (ID#i18-00712_CR3) and is compliant with the Health Insurance Portability and Accountability Act. Informed consent was waived since the study presents no more than minimal risk. This study is reported following the TRIPOD guidelines[53].

**NYU breast ultrasound dataset.** The dataset used in this study was collected from NYU Langone Health system (New York, USA) across 20 imaging sites. The final dataset contained 288,767 exams (5,442,907 images) acquired from 143,203 patients imaged between January 2012 and September 2019. Each US exam included between 4 and 70 images with 18.8 images per exam on average (Supplementary Fig. 3a). The images had an average resolution of $665 \times 603$ pixels in width and height, respectively (Supplementary Fig. 3b). Both B-mode and color Doppler images were included. For each color Doppler image, the color Doppler map was overlaid on the B-mode US image. The AI system processed both B-mode images and color Doppler images in the same way. A summary of the acquisition devices is shown in Supplementary Table 9. Each exam was associated with additional patient metadata as well as a radiology report summarizing the findings. We extracted breast tissue density from the patients' past mammography reports and assigned "unknown" to patients who did not have any mammography exams. Both screening and diagnostic US exams were included. Screening exams are performed for women who have no symptoms or signs of breast cancer while diagnostic US exams can be used to evaluate women who present with symptoms such as a new lump or pain in the breast or can be used to further evaluate abnormalities detected on a screening examination. While screening exams are typically comprehensive and image both breasts, diagnostic US exams vary in terms of how targeted they are, and might image both breasts, one breast, or sometimes just a single lesion. The dataset was filtered as described in the next section. Further details can be found in the technical report[41].

**Filtering of the dataset.** We initially extracted a dataset of 425,506 breast US exams consisting of 8,448,978 images collected from 212,716 unique patients. We then applied a few levels of filtering to obtain the final dataset for training and evaluating the neural network. This entailed the exclusion of exams with invalid patient identifiers, exams collected before 2012, exams collected from patients younger than 16 years of age, duplicate images, exams from non-female patients, and invalid images based on the `ImageType` attribute, which consisted of non-US images such as reports or demographic data screenshots. We further excluded images that were collected during biopsy procedures based on the `PerformedProcedureStepDescription`, `StudyDescription` & `RequestedProcedureDescription` attributes of the image metadata, in that order, images with missing metadata information relating to the type of procedure, images with more than 80% zero pixels, exams with multiple patient identifiers or study dates, exams with an extreme number of images, and exams with missing image laterality.

Patients were then randomly split among training (60%), validation (10%) and test (30%) sets. After splitting, each patient appeared in only one of the training, validation, and test sets. The training set consisted of 3,930,347 images within 209,162 exams collected from 101,493 patients. The validation set consisted of 653,924 images within 34,850 exams collected from 16,707 patients. The test set consisted of 858,636 images within 44,755 exams collected from 25,003 patients. The training set was used to optimize learnable parameters in the models. The validation set was used to tune the hyperparameters and select the best models. The test set was used to evaluate the performance of the models selected using the validation set. We applied additional filtering on the test set as described in the next section.

**Additional filtering of the test set.** To provide a clinically realistic evaluation of the AI system, we additionally refined the test set using the steps summarized in Supplementary Fig. 4. First, we ensured that each non-biopsied exam was followed with a subsequent cancer-negative exam. Non-biopsied patients who had a negative (BI-RADS 1) or benign (BI-RADS 2) US exams were only included in the test set if they did not have any malignant breast pathology found within 0-15 months following their US exam, and had follow up imaging between 6 and 24 months that was also negative or benign (BI-RADS 1–2). Patients who did not undergo biopsy and had probably benign US exams (BI-RADS 3) were included in the test set if they did not have any malignant breast pathology found within 0–15 months following their exam, and met one of two additional criteria: all of their subsequent US exams in the 4–36 months following their initial US exam were BI-RADS 1–2, or they had at least one follow-up US exam at 24–36 months which was evaluated as BI-RADS 1–3.

Next, we refined exams with biopsy-proven benign findings to determine if the pathology results were deemed by the radiologist to be concordant or discordant with the imaging features of the breast lesion. Patients with biopsy reports that confirmed a discordant benign finding were only included in the test set if they received a subsequent biopsy (that was not discordant) or breast surgery within the 6 months following the initial discordant biopsy. Patients with benign discordant biopsies that did not receive subsequent pathological evaluation were excluded.

Lastly, we ensured that exams with pathology-proven cancers contained images of these cancers. Since breast US produces small images which do not comprehensively capture the entire breast, a proportion of patients diagnosed with breast cancer did not have images of the cancer in any of their US images. US exams with a label indicating malignancy and a BI-RADS score of 1–2 were excluded as these exams typically did not contain images of the cancer. Additionally, patients diagnosed with breast cancer who did not have any breast pathology obtained using US-guided biopsy were also excluded, since the majority of patients diagnosed using MRI and stereotactic-guided biopsies had malignancies that were sonographically occult. US exams that received a BI-RADS score of 0, 3, and 6, as well as patients who had breast pathology obtained using multi-modal image guidance (US plus stereotactic and/or MRI guided biopsies) had their cases manually reviewed to confirm that breast cancer was visible on the US exam. Patients who were given a BI-RADS score of 4-5 and had all their breast pathology obtained using US-guided biopsy were presumed to have visible cancers and were not manually reviewed.

**Breast-level cancer labels.** Among all the exams in the dataset, 28,914 exams (approximately 10%) were associated with at least one pathology report dated within 30 days prior or 120 days after the US examination. Pathology reports were used to automatically detect cancer labels. In cases where there were multiple pathology reports recorded within the considered time window, all of these reports were evaluated. Malignant findings included primary breast cancers: invasive ductal carcinoma, invasive lobular carcinoma, special-type invasive carcinoma (including tubular, mucinous and cribriform carcinomas), inflammatory carcinoma, intraductal papillary carcinoma, microinvasive carcinoma, ductal carcinoma in situ, as well as non-primary breast cancers: lymphoma and phyllodes. Benign findings included cyst, fibroadenoma, scar, sclerosing adenosis, lobular carcinoma in situ, columnar cell changes, atypical lobular hyperplasia, atypical ductal hyperplasia, papilloma, periductal mastitis and usual ductal hyperplasia. The labels were automatically extracted from the corresponding pathology reports using a natural language processing pipeline developed earlier[41]. Of note, patients with multiple pathology reports could be assigned both malignant and benign labels if their exam contained both types of lesions.

**Breast ultrasound images dataset.** This external dataset was collected in 2018 from Baheya Hospital for Early Detection and Treatment of Women's Cancer (Cairo, Egypt) with the LOGIQ E9 ultrasound system. It included 780 breast US images, with an average resolution of $500 \times 500$ pixels, acquired from 600 female patients whose ages ranged between 25 and 75 years old. Among these 780 images, 133 were normal images without cancerous masses, 437 were images containing malignant masses and 210 were images with benign masses. We refer the reader to the original paper for more information about this public dataset[40].

**Deep neural network architecture.** We present a deep learning model (DLM) whose architecture is shown in Supplementary Fig. 5. To explain the mechanics of this model, we need to introduce some notations. Let $\mathbf{x} \in \mathbb{R}^{H,W,3}$ denote an RGB US image with a resolution of $H \times W$ pixels and let $\mathbf{X} = \{\mathbf{x}_1, \mathbf{x}_2, \ldots, \mathbf{x}_K\}$ denote an image set that contains all images acquired from the patient during an US exam from one breast. This DLM is trained to process the image set $\mathbf{X}$, which may vary in number of the images it contains (Supplementary Fig. 3), and generate two probability estimates $\hat{y}^b, \hat{y}^m \in [0, 1]$ that indicate the predicted probability of the presence of benign and malignant lesions in the patient's breast, respectively. The DLM is designed to resemble the diagnostic procedure performed by radiologists. First, it generates saliency maps and probability estimates for each image $\mathbf{x}_k$ in the image set. This step is similar to a radiologist roughly scanning through each US image and looking for abnormal findings. Then it computes a set of attentions scores which indicate the importance of each image to the cancer diagnosis task. This procedure can be seen as an analog to a radiologist concentrating on images that contain suspicious lesions. Finally, it forms a breast-level cancer diagnosis by combining information collected from all images. This is analogous to modeling a radiologist comprehensively considering signals in all images to render a full diagnosis. Below we describe each step in detail.

1. *Saliency maps.* The DLM first utilizes a convolutional neural network[54] $f_g$ (parameterized as ResNet-18[55]) to extract a representation of each image $\mathbf{x}_k$, in an image set $\mathbf{X}$, denoted by $\mathbf{h}_k \in \mathbb{R}^{h,w,C}$. The height, the width, and the number of channels are denoted by $h$, $w$, and $C$, respectively. Inspired by Zhou et al.[38], we then apply a convolutional layer with $1 \times 1$ convolutional filters followed by sigmoid non-linearity to transform $\mathbf{h}_k$ into two saliency maps $\mathbf{A}_k^b \in \mathbb{R}^{h,w}$ and $\mathbf{A}_k^m \in \mathbb{R}^{h,w}$. These saliency maps highlight approximate locations of benign and malignant lesions in each image. Each

element $\mathbf{A}_k^b[i,j]$, $\mathbf{A}_k^m[i,j] \in [0,1]$ denotes the contribution of spatial location $(i,j)$ towards predicting the presence of benign/malignant lesions. The resolutions of the saliency maps $(h,w)$ depends on the implementation of $f_g$. The sizes $(h,w)$ are usually smaller than the resolution of the input image $(H,W)$. In this work, we set $h = w = 8$, $C = 512$, and $H = W = 256$.

2. *Attention scores*. The images in the image set $\mathbf{X}$ might significantly differ in how relevant each of them is to the classification task. To address this issue, we utilize the Gated Attention Mechanism[56], allowing the model to select which information to incorporate from all images. Specifically, we first apply global max pooling to transform the representation $\mathbf{h}_k$ computed for the image $\mathbf{x}_k$ into a vector $\mathbf{v}_k \in \mathbb{R}^C$. Two attention scores $\alpha_k^b$ and $\alpha_k^m \in [0,1]$ that indicate the importance of each image $\mathbf{x}_k$ to the estimation of the probability of the presence of benign and malignant findings in the breast are computed as

$$\boldsymbol{\alpha}_k = \frac{\exp\{\mathbf{W}^{\mathsf{T}}(\tanh(\mathbf{V}\mathbf{v}_k^{\mathsf{T}}) \odot \mathrm{sigm}(\mathbf{U}\mathbf{v}_k^{\mathsf{T}}))\}}{\sum_{j=1}^{K} \exp\{\mathbf{W}^{\mathsf{T}}(\tanh(\mathbf{V}\mathbf{v}_j^{\mathsf{T}}) \odot \mathrm{sigm}(\mathbf{U}\mathbf{v}_j^{\mathsf{T}}))\}}, \qquad (1)$$

where $\boldsymbol{\alpha}_k = \begin{bmatrix} \alpha_k^b \\ \alpha_k^m \end{bmatrix}$ denotes the concatenation of attention scores for both benign and malignant findings, $\odot$ denotes an element-wise multiplication, and $\mathbf{W} \in \mathbb{R}^{L,2}$, $\mathbf{V} \in \mathbb{R}^{L \times M}$ and $\mathbf{U} \in \mathbb{R}^{L \times M}$ are matrices of learnable parameters. In all experiments, we set $L = 128$ and $M = 512$.

3. *Cancer diagnosis*. Lastly, the DLM aggregates the information from all US images in the image set $\mathbf{X}$ and generates the final diagnosis using the attention scores and saliency maps. We first use an aggregation function $f_{\mathrm{agg}}(\mathbf{A}) : \mathbb{R}^{h,w} \mapsto [0,1]$ to transform the saliency maps into image-level predictions:

$$\hat{y}_k^b = f_{\mathrm{agg}}(\mathbf{A}_k^b) \quad \hat{y}_k^m = f_{\mathrm{agg}}(\mathbf{A}_k^m). \qquad (2)$$

In our work, we parameterize $f_{\mathrm{agg}}$ as the top $t\%$ pooling proposed by Shen et al.[57–59]. Namely, we define the aggregation function as

$$f_{\mathrm{agg}}(\mathbf{A}) = \frac{1}{|H^+|} \sum_{(i,j) \in H^+} \mathbf{A}_{i,j}, \qquad (3)$$

where $H^+$ denotes the set containing locations of top $t\%$ values in $\mathbf{A}$, and $t$ is a hyperparameter. The breast-level cancer prediction $\hat{\mathbf{y}} = \begin{bmatrix} \hat{y}^b \\ \hat{y}^m \end{bmatrix}$ is then defined as the average of all image-level cancer predictions weighted by the attention scores:

$$\hat{y}^b = \sum_{k=1}^{K} \alpha_k^b \hat{y}_k^b, \quad \hat{y}^m = \sum_{k=1}^{K} \alpha_k^m \hat{y}_k^m. \qquad (4)$$

**Training details**. In order to constrain the saliency maps to only highlight important regions, we impose the $L_1$ regularization on $\mathbf{A}$ which penalizes the DLM for highlighting irrelevant pixels:

$$L_{\mathrm{reg}}(\mathbf{A}) = \sum_{(i,j)} |\mathbf{A}[i,j]|. \qquad (5)$$

Despite the relative complexity of our proposed framework, this DLM can be trained end-to-end using stochastic gradient descent with the following loss function, defined for a single training example (i.e. one breast) as

$$L(\mathbf{y}, \hat{\mathbf{y}}) = \sum_{c \in \{b,m\}} \mathrm{BCE}(y^c, \hat{y}^c) + \beta \sum_{k=1}^{K} L_{\mathrm{reg}}(\mathbf{A}_k^c), \qquad (6)$$

where BCE is the binary cross-entropy and $\beta$ is a hyperparameter. For all experiments, the training loss is optimized using Adam[60]. Of note, labels indicating the presence of benign lesions ($y^b$) were also used during training to regularize the network through multi-task learning[61]. On the test set, we focus on evaluating predictions of malignancy since it is a more clinically relevant task: identification of malignant lesions has an immediate and significant impact on patient management (biopsy, potential surgery), whereas identification of a benign breast lesions typically does not alter management compared to patients without breast lesions[12].

We optimized the hyperparameters with random search[62]. Specifically, we searched for the learning rate $\eta \in 10^{[-5.5,-4]}$ on a logarithmic scale, regularization hyperparameter $\beta \in 10^{[-3,0.5]}$ on a logarithmic scale, weight decay hyperparameter $\lambda \in 10^{[-6,-3.5]}$ on a logarithmic scale, and the pooling threshold $t \in [0.1,0.5]$ on a linear scale. We trained 30 separate models using hyperparameters uniformly sampled from the ranges above. Each model was trained for 50 epochs. We saved the model weights from the training epoch that achieves the highest AUROC on the validation set. To further improve our results, we used model ensembling[63]. Specifically, we average the breast-level predictions of the top 3 models that achieved the highest AUROC on the validation set to produce the overall prediction of the ensemble.

During training, we adopt image augmentation including random horizontal flipping ($p = 0.5$), random rotation ($-45°$ to $45°$), random translation in both horizontal and vertical directions (up to 10% of the image size), scaling by a random factor between 0.7 and 1.5, and random shearing ($-25°$ to $25°$). The resulting image was then resized to $256 \times 256$ pixels using bilinear interpolation

and normalized. During the validation and test stages, the original image was resized and normalized without any augmentation.

**Test-time augmentation**. We adopted test-time augmentation[64] on the external test set to improve model's performance. We applied following augmentations and computed a prediction on each augmented image: random horizontal flipping ($p = 0.5$), random vertical flipping ($p = 0.5$), and altering the brightness and contrast by a factor randomly chosen from [0.9, 1.1]. This augmentation pipeline was selected using AI's performance on the validation subset of the NYU Breast Ultrasound Dataset. We repeated this procedure 20 times on each image. The final prediction for each image was computed by averaging the predictions on all augmented images.

**Implementation details**. Image preprocessing was performed using Python (3.7) with the following packages: OpenCV (3.4), pandas (0.24.1), Numpy (1.15.4), PIL (5.3.0), and Pydicom (2.2.0). Deep learning model was implemented using PyTorch (1.1.0) and Torchvision (0.2.2). Evaluation metrics were computed using Sklearn (0.19.1).

**Reader study**. We performed a reader study to compare the performance of the proposed DLM with breast radiologists. This study included ten board-certified breast radiologists with an average of 15 years of clinical experience (Supplementary Table 2). Their experience ranged from 3 to 40 years. Nine of the ten radiologists were fellowship-trained in breast imaging. The one radiologist who did not receive formal fellowship training (R10) worked as a sub-specialized breast radiologist and had over 30 years of breast imaging experience. The readers were provided with US images including metadata (breast laterality, position of the probe, notes from the sonographer) and the age of the patient. For each breast in all exams, the readers were then asked to provide a diagnostic BI-RADS score using the values 1, 2, 3, 4A, 4B, 4C or 5. A score of 0 was not permitted.

**Hybrid model**. To explore the potential benefit that the AI system might be able to provide, we created a hybrid model for each radiologist, whose predictions were created by averaging the predictions of the respective radiologist and the AI model: $\hat{\mathbf{y}}_{\mathrm{hybrid}} = \lambda \hat{\mathbf{y}}_{\mathrm{expert}} + (1 - \lambda)\hat{\mathbf{y}}_{\mathrm{AI}}$. The BI-RADS scores of radiologists were used as their predictions. Both $\hat{\mathbf{y}}_{\mathrm{AI}}$ and $\hat{\mathbf{y}}_{\mathrm{expert}}$ were standardized to have zero mean and unit variance. In this study, we set $\lambda = 0.5$. We note that $\lambda = 0.5$ is not the optimal value. On the one hand, the performance obtained by retroactively fine-tuning $\lambda$ on the reader study is not transferable to realistic clinical settings. Therefore, we chose $\lambda = 0.5$ as the most natural way of aggregating two predictions without prior knowledge of their quality.

**Statistical analysis**. In this study, we evaluated the performance of the AI system, radiologists, and the hybrid models using the following evaluation metrics: area under receiver operating characteristic curves (AUROC), area under precision-recall curve (AUPRC), sensitivity, specificity, biopsy rate, negative predictive value (NPV), and positive predictive value (PPV). AUROC and AUPRC were used to assess the diagnostic accuracy of the probabilistic predictions generated by the AI system/hybrid models and the BI-RADS scores of the readers. The BI-RADS scores were treated as a 6-point index of suspicion for malignancy: scores of 1 and 2 were collapsed into the lowest category of suspicion; scores 3, 4A, 4B, 4C and 5 were treated independently as increasing levels of suspicion. AUROC avoids the subjectivity in selecting the thresholds to dichotomize continuous predictions, since it compares performance across all possible recall rates. However, AUROC weights omission and commission errors equally and therefore could provide excessively optimistic estimates in extremely imbalanced classification tasks such as cancer diagnosis where the negative cases often overwhelm the positive cases[65]. Therefore, to complement AUROC, we also reported AUPRC which solely evaluates the ability to correctly identify the positive cases. We calculated both AUROC and AUPRC using the Python Scikit-learn API[66].

In addition, we also evaluated the binary predictions of the AI system, the hybrid models, and the readers using sensitivity, specificity, biopsy rate, NPV, and PPV. These metrics are commonly used to assess the diagnostic accuracy in clinical studies[7,11,15]. The PPV reported in this study corresponds to PPV$_2$, which is defined as the number of breasts with cancer that were recommended to undergo biopsy divided by the total number of breast biopsies recommended[12]. For each breast, the AI system and the hybrid models produced a probabilistic score that represents the likelihood of cancer being present. We dichotomized these scores to produce binary predictions by selecting a score threshold that separates positive and negative decisions. To compute sensitivity, we dichotomized the AI system's probabilistic predictions to match average reader's specificity. To calculate the specificity, biopsy rate, PPV and NPV, we dichotomized the AI system's probabilistic predictions by matching average reader's sensitivity. We similarly dichotomized the predictions of each hybrid model using the sensitivity/specificity of its respective reader. For all evaluation metrics, we estimated the confidence intervals at 95% by 1000 iterations of the bootstrap method[67].

In the reader study, we compared the AUROC, AUPRC, sensitivity, specificity, PPV, and biopsy rate of the AI system and hybrid models with those of the average radiologists. The confidence interval for these differences was obtained through

1000 iterations of bootstrap method[67]. The p-values were computed using one-tailed permutation test[68]. In each of 10,000 trials, we randomly swapped the AI/hybrid model's score with one of the comparator reader's score for each case, yielding a reader-AI difference sampled from the null distribution. A one-sided p-value was computed by comparing the observed statistic to the empirical quantiles of the null distribution. We used a statistical significance threshold of 0.05.

**Reporting summary**. Further information on research design is available in the Nature Research Reporting Summary linked to this article.

## Data availability

Data supporting the results demonstrated by this study are available within the main text and the Supplementary Information. The external test dataset used in this study is publicly available at https://scholar.cu.edu.eg/?q=afahmy/pages/dataset. The NYU Breast Ultrasound Dataset was obtained under the NYU Langone Health IRB protocol ID#i18-00712_CR3 from the NYU Langone Health private database for the current study and therefore cannot be made publicly available. We published the following report explaining how the dataset was created for reproducibility: https://cs.nyu.edu/~kgeras/reports/ultrasound_datav1.0.pdf. Although, we cannot make the dataset public, we will evaluate models from other research institutions on the test part of the data set upon request. For any further queries regarding data availability, please contact the corresponding author (k.j.geras@nyu.edu). Requests will be answered within one week.

## Code availability

The neural networks used in our AI system were developed in PyTorch. Code for preprocessing the data and running the inference, including the weights of the neural networks, sufficient to evaluate our system on other datasets, is available for research purposes upon a request made to the corresponding author (k.j.geras@nyu.edu). Requests will be answered within one week. At this point, we are not sharing the code publicly in order not to compromise potential commercialization of our system.

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

## Acknowledgements

The authors would like to thank Mario Videna, Abdul Khaja and Michael Costantino for supporting our computing environment, Benny Huang and Marc Parente for extracting the data, Yizhuo Ma for providing graphical design consultation, and Catriona C. Geras for proofreading the manuscript. We also gratefully acknowledge the support of Nvidia Corporation with the donation of some of the GPUs used in this research. This work was supported in part by grants from the National Institutes of Health (P41EB017183, R21CA225175), the National Science Foundation (1922658), the Gordon and Betty Moore Foundation (9683), the Polish National Agency for Academic Exchange (PPN/IWA/2019/1/00114/U/00001) and NYU Abu Dhabi.

## Author contributions

Y.S., F.E.S. and J.O. are the co-first authors of this paper. Y.S., F.E.S. and K.J.G. designed the experiments with neural networks. Y.S. conducted the experiments with neural networks. Y.S., F.E.S., J.O., J.W., K.K., J.P., N.W. and C.H. built the data preprocessing pipeline. Y.S. and F.E.S. conducted the reader study and analyzed the data. Y.S., F.E.S. and J.O. conducted literature search. Y.S. and J.B. conducted the statistical analysis. J.O., C.H., S.W., A.M., R.E., D.A., C.T., N.S., Y.G., C.C., S.G., J.A., C.L., S.K.S., C.L., R.M., C.M., A.L., B.R., L.M. and L.H. collected the data. L.H. analyzed the results from a clinical perspective. K.J.G. and F.E.S. supervised the execution of all elements of the project. All authors provided critical feedback and helped shape the manuscript.

## Competing interests

The authors declare no competing interests.
