## [Peer Review File · Nature Communications]

Reviewers' Comments:

Reviewer #1:

Remarks to the Author:

Dear authors

Thank you for your thorough efforts to revise the paper on this ambitious study using extensive data. The utility of AI in ultrasound interpretation remains to be tested in a clinical setting. I have no further comments or suggestions of improvement.

Reviewer #2:

Remarks to the Author:

Nature Communications (previously submitted to Nature Medicine)

REVIEWER #2

Previous comment #5: At a few places in your responses and specifically here, you did not perform the suggested changes and returned the final decision to the Editor. I strongly believe it would be a fair clarification to specify in the title that the study was retrospective and based on B-mode imaging alone. The Editor should be aware of this recent manuscript where the study design was made clear in the title to avoid any ambiguity or to emphasize, in this case, the strength of the work: see "Prospective assessment of breast cancer risk from multimodal multiview ultrasound images via clinically applicable deep learning, *Nature BME*, 2021". In your case, this reviewer appreciates the change made in the abstract to clarify the retrospective nature of the work: "In a retrospective reader study". Please clarify in the title that your study is based on B-mode imaging. To emphasize a strength of your study, you may add in the title the number of cases considered in your AI model.

Previous comment #8: The improvement in performance over radiologist's BI-RADS assessment can be found, for example, in the paper by Destremes mentioned in the original review of this manuscript in *Nature Medicine*: Destremes et al., *UMB*, 46, 2, 436-444, 2020. In the latter paper, BI-RADS classification by radiologists provided an AUC of 95% that could be improved by 2% by adding other ultrasound modalities than B-mode in the machine learning model. The authors are right, I do not know a paper on comparative performance of B-mode BI-RADS versus B-mode AI.

Previous comment #9: I understand that this is a strength of your study but clarify into the manuscript inclusion and exclusion criteria. Also, I understand that the wording in the machine learning field may differ than the medical field. However, your targeted audience is also radiologists. I would strongly suggest early in the manuscript to clarify that the meaning of "prediction" may differ than "medical prediction" that would require a prospective design.

Other issue (NYU breast ultrasound dataset): In response to one of my comment you added this: "Both B-mode and color Doppler images were included". However, I am not sure on how this was considered in your AI algorithm. Did you also analyse color Doppler maps and/or pulse-wave Doppler spectrograms, probably not? You may have taken B-mode DICOM images extracted from the duplex color Doppler exam. This needs to be clarified because you likely did not include flow velocity information into the AI design.

Overall, this reviewer is satisfied by other responses provided and by the revision of the manuscript.

Reviewer #4:

Remarks to the Author:

Dear Editor,

The authors did a thorough revision and addressed all my concerns satisfactory.

This is an interesting topical paper and I would very much like to see this published.

I would advocate to include the results of the small reader study of the external data set in the final paper as further highlights the benefits of an AI-decision support system.

Best Katja

Reviewer #5:

Remarks to the Author:

This manuscript developed and validated, both internally and externally, an AI system for breast cancer diagnosis using ultrasound images. A large database was used to train the AI model. The performance of the proposed AI model was assessed and compared with radiologists.

My first major concern is the novelty of the AI technique. The Resnet has been used as a standard backbone network for years. This Resnet-based AI system achieved good performance on the datasets used in this article. It would be welcome to design some task-specific CNN or other innovative learning techniques, especially for such a large database.

My second major concern is the generalization assessment. The model was developed using a large dataset but tested in only one external dataset from one single center. The ultrasound data in this external set were acquired using a single device, the GE logiqE9. I guess the AI system built using such a large dataset might be powerful. But the lack of multicenter evidence is a clear drawback.

The authors reported a rather low radiologist's AUROC on the external validation set. Only one radiologist did the test, so bias might be induced, for example the radiologist might not very familiar with the ultrasound data produced by logiqE9. It would be better to ask more radiologists to repeat the test on the same cases.

The authors claimed that "the AI system was trained to perform classification and localization in a weakly supervised manner". Generally, there are three types of weakly supervised learning: incomplete supervised learning where only part of the training data is labelled; inaccurate supervised learning where some labels are noised or wrong; and inexact supervised learning where the labels are coarse-grained. Here I guess the authors referred to the third type - learning to locate lesions with coarse-grained image-level labels (malignant or benign). to localize the lesion with coarse-grained image-level labels (malignant or benign). However, the AI model was a classical classification model (ResNet), rather than a localization model or segmentation model. The localization was done with only a heatmap-like saliency map, rather than generating a bounding-box containing a specific object. In other words, the model was designed for binary classification, and the labels were complete, accurate and exact. No localization subnet was presented. I suggest the author not use the term "weakly supervised", as it may mislead the readers.

The authors also claimed that "an interpretable AI system trained with such a large dataset could help discover novel data-driven imaging biomarkers, leading to a better understanding of breast cancer". I suggest that the authors use the term 'interpretable AI' with caution. The saliency map only provides a possible technique to show where the AI model focuses on to make a prediction. It does not explain what it learns from images, let alone what human-understandable meanings the learned features have. In other word, the AI model is still a black box. Telling where the AI focused on does not directly means an interpretable AI. Except 'where', 'what' and 'how' are more

important.

It would be interesting to calculate the class activation map (CAM) from the CNN, as it provides another way to show where the AI model pays attention. CAM tells which regions contribute more to the final prediction. Comparing CAM with the saliency map used by the author will draw some useful conclusions - which of these two technologies is more beneficial for understanding the presented AI model.

The saliency map given by the author is too few. It would be required to show more saliency maps, preferably some more unusual saliency maps for human experts. For example, are the peritumor areas useful for identifying breast cancer? Which imaging patterns are the most discriminative?

Response to Reviewers

We are glad that all reviewers appreciated our effort and accepted our answers. Reviewer 1 and Reviewer 3 deemed the previous revision satisfactory. Reviewer 2 and Reviewer 5 provided additional constructive comments. In this revision, we addressed their comments and highlighted our updates in the revised version of the manuscript. Below, we respond to all specific points made by Reviewer 2 and Reviewer 5.

Reviewer 2 suggested adding a few additional elements to the title: (1) highlighting that the study is retrospective, (2) adding the information that we specifically use B-mode and Color Doppler images (which is the standard of care for breast imaging in the United States) and (3) information on the number of images used during training. While, we think these are fair points and we highlighted these elements to the abstract, we feel that the suggested changes would make the title, which is already long (12 words, 110 characters), extremely long (>20 words, >200 characters). We tend to think that the title is fine as it is and that the abstract is very clear. However, we would like to defer a decision on whether we should indeed incorporate these elements in the title to the editor.

Reviewer 2

Previous Comment 8 *The improvement in performance over radiologist’s BI-RADS assessment can be found, for example, in the paper by Destrempes mentioned in the original review of this manuscript in Nature Medicine: Destrempes et al., UMB, 46, 2, 436-444, 2020. In the latter paper, BI-RADS classification by radiologists provided an AUC of 95% that could be improved by 2% by adding other ultrasound modalities than B-mode in the machine learning model. The authors are right, I do not know a paper on comparative performance of B-mode BI-RADS versus B-mode AI.*

We added the suggested references [2, 7] to the introduction and discussion sections.

Previous Comment 9a *I understand that this is a strength of your study but clarify into the manuscript inclusion and exclusion criteria.*

We provided a detailed clarification on the inclusion and exclusion criteria in the Methods section (see “Filtering of the dataset” and “Additional filtering of the test set”). We also illustrated the filtering workflow utilized in this work in Supplementary Figure 4.

Previous Comment 9b *Also, I understand that the wording in the machine learning field may differ than the medical field. However, your targeted audience is also radiologists. I would strongly suggest early in the manuscript to clarify that the meaning of “prediction” may differ than “medical prediction” that would require a prospective design.*

We followed the reviewer’s suggestion and added the following clarification in the last paragraph of page 3 where the term “prediction” appears the first time in the manuscript: *Of note, the term “prediction” refers to the diagnosis produced by AI/radiologists in this retrospective study as it is often used in the machine learning literature. It does not imply the study being prospective.*

Comment 1 *In response to one of my comment you added this: “Both B-mode and color Doppler images were included”. However, I am not sure on how this was considered in your AI algorithm. Did you also analyse color Doppler maps and/or pulse-wave Doppler spectrograms, probably not? You may have taken B-mode DICOM images extracted from the duplex color Doppler exam. This needs to be clarified because you likely did not include flow velocity information into the AI design.*

The AI system processed both B-mode images and color Doppler images in the same way. For each color Doppler image, the color Doppler map was overlaid on the B-mode US image. Both B-mode images and color Doppler images were given to the network, which extracted representations from them and combined them using an attention mechanism such that a prediction on an exam level was made. Therefore, although the AI system was not designed to separately analyze blood flow velocity, such information was available to the AI and was integrated along with other visual features for diagnosis. We updated the Methods section to provide further clarification on this comment.

Reviewer 5

Comment 1 *My first major concern is the novelty of the AI technique. The Resnet has been used as a standard backbone network for years. This Resnet-based AI system achieved good performance on the datasets used in this article. It would be welcome to design some task-specific CNN or other innovative learning techniques, especially for such a large database.*

We would like to clarify that although ResNet is an element of the proposed AI system, the entire system is significantly different from a canonical ResNet. It consists of several components that are designed specifically for the breast ultrasound interpretation task. We explain these task-specific designs below in detail.

First, while traditional ResNets need to be trained with image-level cancer labels (presence or absence of cancer in each image), the neural network proposed in this work is able to learn from breast-level labels (i.e. one label per breast exam, which might contain multiple images, rather than one label per image). To enable this, we designed a novel attention-based aggregator that learns from breast-level labels which can be automatically extracted from pathology reports. This design is particularly important since the pathology reports only confirm the presence of benign/malignant lesions in a breast but do not specify in which ultrasound images these lesions are visible. Collecting image-level cancer labels requires human experts to manually identify images that contain lesions and is impractical for large-scale datasets such as the one used in this work. The technical details of this design are in the Methods section (see “Deep neural network architecture”).

Second, our model provides an explanation of its predictions using saliency maps, generated as a byproduct of the inference process, which highlight the regions in the ultrasound image that the model deems important. This differentiates it from traditional neural-network-based classifiers (such as ResNets for example) which only provide “black-box” predictions. To achieve that, we built upon architecture designs from the weakly supervised object localization literature [6, 11, 12] that enable the model to produce such saliency maps using only the multiple instance learning paradigm. The saliency maps produced by our network are not generated post hoc based on gradient backpropagation [5, 9, 10]. Our network is also different from classical object segmentation/detection models, such as U-Net [8] or Mask R-CNN [3], which need to be trained with expensive pixel-level segmentation/bounding box labels. Our design effectively improves interpretability of our model while avoiding the cost of additional labeling efforts.

Comment 2 *My second major concern is the generalization assessment. The model was developed using a large dataset but tested in only one external dataset from one single center. The ultrasound data in this external set were acquired using a single device, the GE logiqE9. I guess the AI system built using such a large dataset might be powerful. But the lack of multicenter evidence is a clear drawback. The authors reported a rather low radiologist’s AUROC on the external validation set. Only one radiologist did the test, so bias might be induced, for example the radiologist might not very familiar with the ultrasound data produced by*

logiqE9. It would be better to ask more radiologists to repeat the test on the same cases.

We would like to clarify that NYU Langone Health is an academic health center that comprises of multiple affiliated hospitals and operates over 350 sites. The NYU Breast Ultrasound Dataset, collected at this academic center, includes a large patient population (143,203 patients) and a diverse range of ultrasound devices (20 models), acquired over 20 imaging sites.

We agree with the reviewer that the external test set has limitations in representativeness. On the other hand, we also acknowledge that this is the largest breast ultrasound dataset that is publicly available. We followed the reviewer’s suggestion and conducted an additional reader study with two attending radiologists (reader B: 20 years of experience and fellowship-trained, reader C: 11 years of experience and fellowship-trained) on the external test set. Including reader A from the previous revision, the three radiologists achieved an average AUROC of 0.888 (SD: 0.004, 95% CI: 0.855, 0.909). This is a significantly lower score than our model’s (0.927 95% CI: 0.907, 0.959). We summarized the results from this additional reader study in Supplementary Table 10.

Comment 3 *The authors claimed that “the AI system was trained to perform classification and localization in a weakly supervised manner”. Generally, there are three types of weakly supervised learning: incomplete supervised learning where only part of the training data is labelled; inaccurate supervised learning where some labels are noised or wrong; and inexact supervised learning where the labels are coarse-grained. Here I guess the authors referred to the third type - learning to locate lesions with coarse-grained image-level labels (malignant or benign). to localize the lesion with coarse-grained image-level labels (malignant or benign). However, the AI model was a classical classification model (ResNet), rather than a localization model or segmentation model. The localization was done with only a heatmap-like saliency map, rather than generating a bounding-box containing a specific object. In other words, the model was designed for binary classification, and the labels were complete, accurate and exact. No localization subnet was presented. I suggest the author not use the term “weakly supervised”, as it may mislead the readers.*

We agree that this point deserves a clarification. We adopted reviewer’s suggestion and changed the sentence in paragraph 4 of page 3 to “*In addition to classifying the images, the AI system also localizes the lesions in a weakly supervised manner [6, 11, 12].*”.

While our model was trained with with coarse-grained labels (breast-level labels), it has the ability to localize suspected malignant and benign lesions through the layer in the network that generates the saliency maps. Even though we did not produce bounding box predictions based on these saliency maps, it is technically straight-forward to do so. To clarify this point, we summarize the workflow of our model below.

The AI system first uses ResNet-18 to extract a feature representation from each ultrasound image in an exam. It then converts feature representations to saliency maps using a layer with one by one convolution and sigmoid non-linearity. Next, these saliency maps were collapsed into image-level predictions using a pooling layer. Finally, the model employs an attention-based aggregator to produce a breast-level cancer diagnosis. Although our model does not have a dedicated localization sub-net, it has a dedicated layer, which localizes the lesions that contribute to its classification prediction. We would like to highlight that the saliency maps produced by our model are not based gradient-backpropagation methods [9, 1]. A detailed technical description of this neural network is in the Methods section (see “Deep neural network architecture” and “Training details”).

Comment 4 *I suggest that the authors use the term ‘interpretable AI’ with caution. The saliency map only provides a possible technique to show where the AI model focuses on to make a prediction. It does not explain what it learns from images, let alone what human-understandable meanings the learned features have. In other word, the AI model is still a black box. Telling where the AI focused on does not directly means an interpretable AI. Except ‘where’, ‘what’ and ‘how’ are more important.*

We agree that “interpretable AI” is a term that requires caution. As there is no consensus what interpretable AI even is, we tried to use it in a way that is consistent with how it is typically used in the machine learning community and in a way that does not overstate what we have done. We made edits in the Discussion section to highlight this point: *Admittedly, the literature has not yet reached a consensus on the definition of what exactly interpretability for neural networks is. Nevertheless, with the saliency maps, researchers can perform qualitative error analysis and understand the strength and limitations of the AI system..*

Comment 5 *It would be interesting to calculate the class activation map (CAM) from the CNN, as it provides another way to show where the AI model pays attention. CAM tells which regions contribute more to the final prediction. Comparing CAM with the saliency map used by the author will draw some useful conclusions - which of these two technologies is more beneficial for understanding the presented AI model.*

The saliency maps produced by our model are equivalent to CAMs in binary classification. In the paper where CAM was proposed [11], the CAM $M_c \in \mathbb{R}^{H,W}$ for a class $c \in \mathcal{C}$ is defined as (equation 2 in the paper):

$$M_c[x, y] = \sum_{k=1}^K \mathbf{w}_k^c \mathbf{h}^c[x, y], \quad (1)$$

where H, W, K denote the height, width, and number of channels of the feature map \mathbf{h} produced by the last convolution layer and $\mathbf{w} \in \mathbb{R}^{K,|\mathcal{C}|}$ are learnable parameters. For visualization purpose, post-softmax CAMs were often used :

$$\tilde{M}_c[x, y] = \text{softmax}(M_c[x, y]) = \frac{\exp(M_c[x, y])}{\sum_{c' \in \mathcal{C}} \exp(M_{c'}[x, y])}. \quad (2)$$

Similarly, in this manuscript, the saliency map is obtained by applying a 1 by 1 convolution with sigmoid non-linearity on \mathbf{h} :

$$\mathbf{A}[x, y] = \sigma\left(\sum_{k=1}^K \mathbf{w}_k \mathbf{h}[x, y]\right), \quad (3)$$

where $\mathbf{w} \in \mathbb{R}^K$ is the learnable parameters in the kernel of the 1 by 1 convolution layer. The saliency maps (equation 3) is equivalent to CAM (equation 2) under a binary classification formulation where the sigmoid is used to replace softmax.

Comment 6 *The saliency map given by the author is too few. It would be required to show more saliency maps, preferably some more unusual saliency maps for human experts. For example, are the peritumor areas useful for identifying breast cancer? Which imaging patterns are the most discriminative?*

We followed the reviewer’s suggestion and added Supplementary Figure 2 to provide additional visualization of saliency maps. We agree with the reviewer that a large-scale qualitative analysis of the saliency maps could help us understand what the AI system learned exactly. We note that providing deeper insights on differences between the AI and radiologists in interpreting US images requires a systematic study at a much larger scale. We have already completed a related study on AI for screening mammography [4]. However, US has different characteristics and would require conducting an analogous study, which is beyond the scope of this paper. We acknowledged that in the discussion section.

References

- [1] CHATTOPADHAY, A., SARKAR, A., HOWLADER, P., AND BALASUBRAMANIAN, V. N. Grad-cam++: Generalized gradient-based visual explanations for deep convolutional networks. In *2018 IEEE winter conference on applications of computer vision (WACV)* (2018), IEEE, pp. 839–847.

- [2] DESTREMPES, F., TROP, I., ALLARD, L., CHAYER, B., GARCIA-DUITAMA, J., EL KHOURY, M., LALONDE, L., AND CLOUTIER, G. Added value of quantitative ultrasound and machine learning in bi-rads 4–5 assessment of solid breast lesions. *Ultrasound in medicine & biology* 46, 2 (2020), 436–444.
- [3] HE, K., GKIOXARI, G., DOLLÁR, P., AND GIRSHICK, R. Mask r-cnn. In *Proceedings of the IEEE international conference on computer vision* (2017), pp. 2961–2969.
- [4] MAKINO, T., JASTRZEBSKI, S., OLESZKIEWICZ, W., CHACKO, C., EHRENPREIS, R., SAMREEN, N., CHHOR, C., KIM, E., LEE, J., PYSARENKO, K., ET AL. Differences between human and machine perception in medical diagnosis. *arXiv preprint arXiv:2011.14036* (2020).
- [5] MONTAVON, G., BINDER, A., LAPUSCHKIN, S., SAMEK, W., AND MÜLLER, K.-R. Layer-wise relevance propagation: an overview. *Explainable AI: interpreting, explaining and visualizing deep learning* (2019), 193–209.
- [6] OQUAB, M., BOTTOU, L., LAPTEV, I., AND SIVIC, J. Is object localization for free?-weakly-supervised learning with convolutional neural networks. In *Proceedings of the IEEE Conference on Computer Vision and Pattern Recognition* (2015), pp. 685–694.
- [7] QIAN, X., PEI, J., ZHENG, H., XIE, X., YAN, L., ZHANG, H., HAN, C., GAO, X., ZHANG, H., ZHENG, W., ET AL. Prospective assessment of breast cancer risk from multimodal multiview ultrasound images via clinically applicable deep learning. *Nature Biomedical Engineering* 5, 6 (2021), 522–532.
- [8] RONNEBERGER, O., FISCHER, P., AND BROX, T. U-net: Convolutional networks for biomedical image segmentation. In *International Conference on Medical image computing and computer-assisted intervention* (2015), Springer, pp. 234–241.
- [9] SELVARAJU, R. R., COGSWELL, M., DAS, A., VEDANTAM, R., PARIKH, D., AND BATRA, D. Grad-cam: Visual explanations from deep networks via gradient-based localization. In *Proceedings of the IEEE international conference on computer vision* (2017), pp. 618–626.
- [10] SUNDARARAJAN, M., TALY, A., AND YAN, Q. Axiomatic attribution for deep networks. In *International Conference on Machine Learning* (2017), PMLR, pp. 3319–3328.
- [11] ZHOU, B., KHOSLA, A., LAPEDRIZA, A., OLIVA, A., AND TORRALBA, A. Learning deep features for discriminative localization. In *Proceedings of the IEEE Conference on Computer Vision and Pattern Recognition* (2016), pp. 2921–2929.
- [12] ZHOU, Z.-H. A brief introduction to weakly supervised learning. *National Science Review* 5, 1 (2018), 44–53.